# Teacher Mindframes from an Educational Science Perspective

**Klaus Zierer \*, Christina Lachner \*, Jonas Tögel and Denise Weckend**

Department of School Education, University of Augsburg, 86159 Augsburg, Germany;
jonas.toegel@phil.uni-augsburg.de (J.T.); denise.weckend@phil.uni-augsburg.de (D.W.)

\* Correspondence: klaus.zierer@phil.uni-augsburg.de (K.Z.); christina.lachner@phil.uni-augsburg.de (C.L.)

**Abstract:** In this article, we describe the philosophical and the scientific background of teacher mindframes and we argue that educational professionalism consists not only of ability and knowledge (competence), but also of will and judgement (attitudes). To back up our argument, we present the results of our current research project on this matter.

**Keywords:** teacher mindframes; educational expertise; integral philosophy

## 1. Introduction

The argumentation in this paper focuses on an insight from Johann Friedrich Herbart's *Outlines of Educational Doctrine* from the year 1835: "Man's worth does not, it is true, lie in his knowing, but in his willing. But there is no such thing as an independent faculty of will. Volition has its roots in thought" [1] (p. 40). This interdependence between knowledge and will appears to have been forgotten in the current educational science discussion, particularly in regard to teacher education. And yet it is of fundamental importance, in view of the latest findings from empirical educational research, and must be brought back into the discussion, especially as far as teacher attitudes are concerned. That is the objective of this article.

To this end, we begin by describing the current state of research on teacher professionalism, through which it becomes evident that studies often focus on subject matter knowledge in the shape of knowledge and ability—without, however, providing any evidence that it is essential for professionalism. Even more recent studies demonstrating that subject matter knowledge must always be accompanied by pedagogical and didactic competence (e.g., [2,3]) fail to recognize that the effectiveness of this triad depends primarily on the attitudes of the teacher [4,5]. John Hattie uses this construct to draw attention away from knowledge and ability towards will and judgment. These thoughts on the current state of research thus lead to an epistemological reflection we would like to approach with the help of Ken Wilber's integral philosophy [6]. Wilber makes it clear with his quadrant model that complex phenomena may be considered from an exterior perspective or an interior perspective, that these perspectives are dependent on each other, and that they are therefore both important. As we demonstrate, knowledge and ability may be classified as belonging to the exterior perspective and will and judgment to the interior perspective. The quadrant model may thus be developed further into the so-called ACAC model (attitude, competence, action, context; c.f. Figure 5) and placed at the center of educational expertise [7]. This model is currently being investigated in several research projects. We conclude the paper by taking a closer look at the results obtained so far.

## 2. On the State of Research: Educational Professionalism as Knowledge and Ability?

There has been much debate in recent years on what constitutes teacher professionalization and how to develop and foster it, both within the scientific community as well as publicly (cf. [8–11])

and in the public sphere (cf. [12,13]). The starting point for this debate was a series of large-scale comparative studies, such as PISA (Programme for International Student Assessment), TIMSS (Trends in International Mathematics and Science Study), and PIRLS (Progress in International Reading Literacy Study), which, while not providing a direct answer to the question of teacher professionalism, do frequently refer to the issue. It is a central issue in several studies, such as TEDS-M, COACTIV, and the IQB state comparison (German Institute for Quality Development in Education). TEDS-M (teacher education development study in Mathematics) is an international comparative study on the effectiveness of teacher education in the subject of mathematics, conducted as part of the DFG-funded (German Research Foundation) project "TED-Unterricht" ("TED-teaching"). The purpose of the IQB state comparative studies commissioned by the Standing Conference of the State Ministers of Education and Cultural Affairs is to determine the extent to which school students in Germany achieve the educational standards that are binding for all states and to identify the areas in which there is a need for additional guidance. The surveys of learning levels are conducted every five years at the primary level in the subjects German and Mathematics and every three years at secondary level I, alternating between the subject groups German, English, and French and Mathematics, Biology, Chemistry, and Physics. The quantity of the data collected for these studies is impressive, and it would seem that they refer to an area of educational science that has been the topic of extensive research. This impression is strengthened by the numerous survey articles on the topic that have appeared in recent years (e.g., [14–16]). It therefore comes as no surprise that there is a consensus concerning the results and the conclusions drawn from them.

We would like to present the IQB land comparison in the following by way of example. Among other things, it investigated whether there is a connection between subject-specific teacher qualifications and student competence acquisition, for instance, whether mathematics and natural science competences of learners vary depending on teacher characteristics regarding qualifications (e.g., teachers who are or are not qualified to teach the subject in question), further vocational training, and demography. The results are clear:

Taking into account all of the aforementioned characteristics, students taught by a Mathematics teacher without qualifications to teach that subject achieved an average of 18 points (SE = 8.5 points) less than those taught by a teacher with qualifications to teach mathematics. On the whole, there were significant correlations between subject-specific teaching qualifications and student competences in three of the four competence areas included in the study, especially at non-academic track secondary schools [11].

Many people regard this as proof that success in school depends critically on the subject matter knowledge of the teacher. Against this background, it must be irritating that John Hattie calculates only a very small effect size of 0.10 for teacher subject matter knowledge in his study "Visible Learning" [4] (p. 300) (cf. Figure 1).

How can this be? The explanation that can be provided following the example of John Hattie serves to resolve this apparent contradiction: Everybody knows teachers who know a lot, but are unable to pass on their knowledge. These teachers lack didactic competence, which involves the ability to illustrate content clearly and explain things well, to point out the most important aspects, and—in even more concrete terms—to create clearly arranged and helpful blackboard drawings and worksheets. Everybody also knows teachers who know an awful lot, but are so unapproachable that they are unable to build a relationship with the learners. They lack pedagogical competence, which involves the ability to establish contact with the students and to create an atmosphere of security, confidence, and trust. Subject matter knowledge on its own, hence, does not lead to an increase in academic achievement. It needs to be accompanied by pedagogical and didactic competence [5].

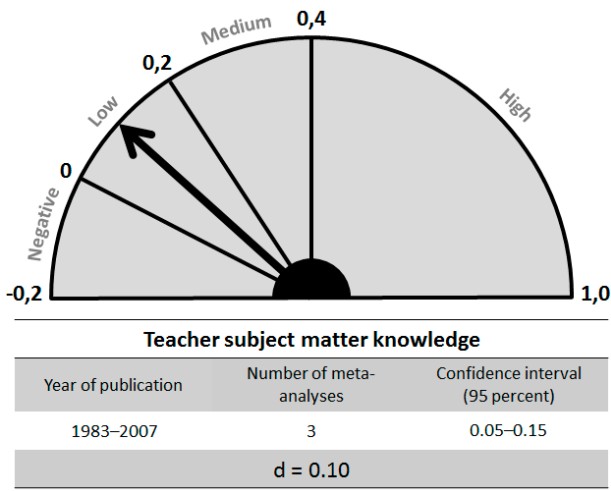

**Figure 1.** The factor "Teacher subject matter knowledge".

This is essentially also the position on this issue in the discourse of the German-speaking world, for instance, in the aforementioned IQB land comparison. After all, teachers who are qualified to teach a particular subject acquired more than just subject matter knowledge during their education; they also acquired subject-specific didactic knowledge and pedagogical knowledge. It is important, at this point, to note that it is not a matter of preferring one of these types of competence over the other two, nor is it a matter of which of them should be given more emphasis. What is crucial, rather, is that pedagogical and didactic competence should be linked. And subject matter knowledge certainly occupies a prominent position in this triad—yet only in combination with the other two. By itself and isolated from the other competence domains, subject matter knowledge is ineffective.

To summarize our reflections up to this point, we can state that knowledge and ability in a particular subject, in didactics, and in pedagogy are sufficient conditions for possessing educational expertise. This is the position advocated, for example, by Jürgen Baumert and Mareike Kunter in their general competence model for teachers [8] (p. 470), which provided the premise for the large-scale comparative studies mentioned above. Yet already in the introduction to this paper, we pointed out with reference to Johann Friedrich Herbart that the value of a human being is not limited to knowledge and ability, but also includes will and judgment and that these modes of being interact. As a consequence, the attempt to take knowledge and ability as the main basis for teacher professionalism must necessarily be a reductionist endeavor and runs the risk of doing justice neither to the human being behind the teacher nor to the pedagogical challenge of teaching and learning. Educational expertise—the term is intended as a means of getting past the historically burdened discourse concerning the correct understanding of teacher professionalism [for more on this issue, cf. [17] (p. 793)—manifests itself not just in knowledge and ability, but also and especially in will and judgment. Empirical evidence for this notion may be found, for instance, in the MET project conducted by the Bill and Melinda Gates Foundation [18], as well as in the works of John Hattie, who emphasizes repeatedly that it is the passionate and enthusiastic teachers [4] that are the key factor in educational processes. He ascribes to them the following mindframes:

1. I focus on learning and the language of learning;
2. I strive for challenge and not merely "doing your best";
3. I recognize that learning is hard work;
4. I built relationships and trust so that learning can occur in a place where it is safe to make mistakes and learn from others;
5. I engage as much in dialogue as monologue;
6. I inform all about the language of learning;
7. I am a change agent and believe that all students can improve;

8.   I give and help students to understand feedback and I interpret and act on feedback given to me;
9.   I see assessment as informing my impact and next steps;
10.  I collaborate with other teachers. [5]

This argumentation leads to the conclusion that it is oftentimes not so important what (knowledge and ability) teachers do in pedagogical contexts, but rather how (will) and why (judgment) they do something.

Interestingly, these findings from empirical educational research can be embedded in an epistemological framework. In the following, we present the integral philosophy of Ken Wilber as an example and then tie it into the argumentation presented so far [6].

## 3. Educational Expertise: Competence and Attitudes

Ken Wilber, currently one of the world's most frequently translated thinkers, drew on ideas by Karl Popper and Jürgen Habermas in developing his epistemology. Its main message is that complex phenomena may be observed from different perspectives and that each and every one of these perspectives is important on its own. He accordingly regards it as problematic to argue from a single perspective. What Ken Wilber thus does is essentially to differentiate between four approaches and assign them to the following model, which he calls the quadrant model. The differences between the approaches may be attributed on the one hand to different means of knowledge acquisition, from interpretation to empirical inquiry, and on the other hand to the difference between the individual and the collective. He therefore proceeds, in accordance with philosophical traditions, from an exterior perspective to an interior perspective and attempts to bring these two worlds together (cf. Figure 2). His undertaking is, hence, nothing less than an attempt to connect all major philosophical currents of the East and the West (including Plotinus, Meister Eckhart, Sri Aurobindo, German Idealism, Advaita Vedanta Hinduism, Tibetan Buddhism, Jean Gebser, Jürgen Habermas, Jean Piaget, Lawrence Kohlberg, Arthur Koestler, Teilhard de Chardin, Alfred North Whitehead, Clare W. Graves, Rupert Sheldrake, and Jiddu Krishnamurti).

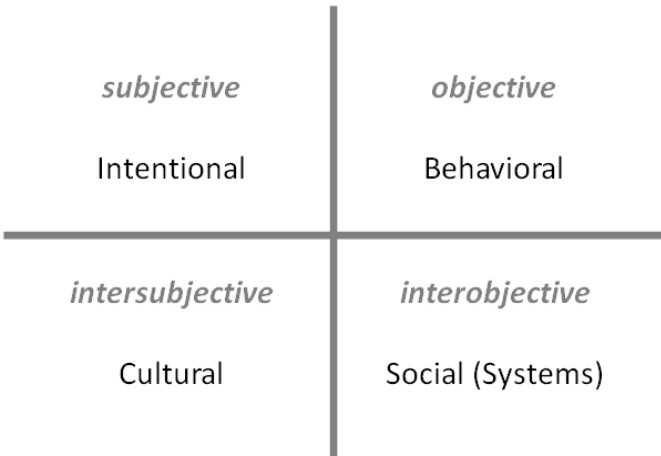

**Figure 2.** Ken Wilber's quadrant model.

Ken Wilber begins by distinguishing an *objective approach* dominated by empirical methods. It achieves a gain in knowledge by means of measurements, tests, and the like. An example of a statement in this quadrant might be "It's raining outside." This statement can be verified quickly and easily by anyone. It is therefore clear that statements taking the objective approach make truth claims. This approach is currently the dominant one in nearly all societal and scientific domains. When we speak of *knowledge* today, what we generally mean is knowledge that may be classified as belonging to this quadrant.

Second, Ken Wilber names a *subjective approach*. This approach is primarily about needs, interests, and feelings. An example would be "I'm fine" as a response to the question "How are you?" The truth content of this statement clearly defies an empirical approach: It is not possible for means of measurements and tests to verify whether the person providing response is telling the truth or lying. We can try to gain additional information by studying the person's facial expression and gestures, but ultimately we are forced to remain at the level of interpretation: We can interpret and attempt to understand how much truth there is in such a statement, but we can never be absolutely certain that our interpretation is correct. Ken Wilber hence posits that statements belonging to the subjective quadrants make claims not to truth, but to truthfulness. These arguments are also particularly true of human *will*. It also defies measurement and testing, but is crucial for thought and action.

The third quadrant in Ken Wilber's model is an *intersubjective approach*. Its key elements are the values, norms, rules, and rituals that influence how people think and act. They can neither be defined empirically nor prescribed by an individual. Rather, they need to be examined in an argumentative and discursive process if they are to attain general acceptance in society. In this respect, statements from the intersubjective approach do not make a claim to truth or to truthfulness, but rather to what Ken Wilber terms cultural fit. As an illustration, consider what values a person's thoughts and actions are determined by and where these values come from. It is not the individual alone who decides what is important and what is not, what is culturally appropriate and what is not; rather, the individual is influenced greatly in his or her *values* by the collective, which may be illustrated by the influence of family in the first years of human life.

Fourth and last, Ken Wilber speaks of an *interobjective approach*, thus taking up systemic relationships: No person exists for him or herself alone, but is integrated into various contexts—into family, into an economic and political system, and into a church, to name what are perhaps the most important ones. According to the systems theory of Niklas Luhmann, which can be associated with this approach, there are numerous points of tension between the individual systems [19]. They may be attributed above all to the different codes with which these systems communicate and work: Politics is primarily a matter of power, economy a matter of profit, church a matter of faith, etc. These varying interests can spark conflicts and controversies, and the individual is called upon to overcome these tensions by means of thought and action. It is above all *ability* that determines whether the individual is capable of forming the different role expectations into a coherent whole. It is therefore a matter of achieving a functional fit, as Ken Wilber terms it.

In considering a complex phenomenon with reference to the quadrant model, it becomes evident that one must differentiate between at least four perspectives. Each of these perspectives is important and cannot be replaced by any of the others. The danger pointed out by Ken Wilber hence lies in arguing in a simplified—and thus reductionist—manner from just one perspective. This necessarily leads to incorrect assumptions and fallacies. As a consequence, Wilber calls for a holistic, one might also say eclectic, consideration of complex phenomena that takes into account at least these four approaches [20].

If we now combine the thoughts on teacher professionalism developed above and the modes of being of human existence derived from them with Ken Wilber's quadrant model, we can make the following classification: Educational expertise is a complex phenomenon. It may be viewed from an interobjective, objective, subjective, and intersubjective perspective. The aspects that stand out are the teacher's ability (interobjective), knowledge (objective), will (subjective), and judgment (intersubjective) with regard to the subject, the didactics, and the pedagogy. Whereas knowledge and ability are commonly associated with subject matter knowledge, didactic competence, and pedagogical competence, will and judgments make up the core of subject-related, didactic, and pedagogical attitudes. All four modes of being of human existence interact with each other and ultimately form the basis of educational expertise. What this means for teacher education is that it should not limit itself to fostering knowledge and ability, but must also focus on the will and the judgments of prospective and active teachers. Figure 3 attempts to illustrate this argumentation in the form of the ACAC model [7].

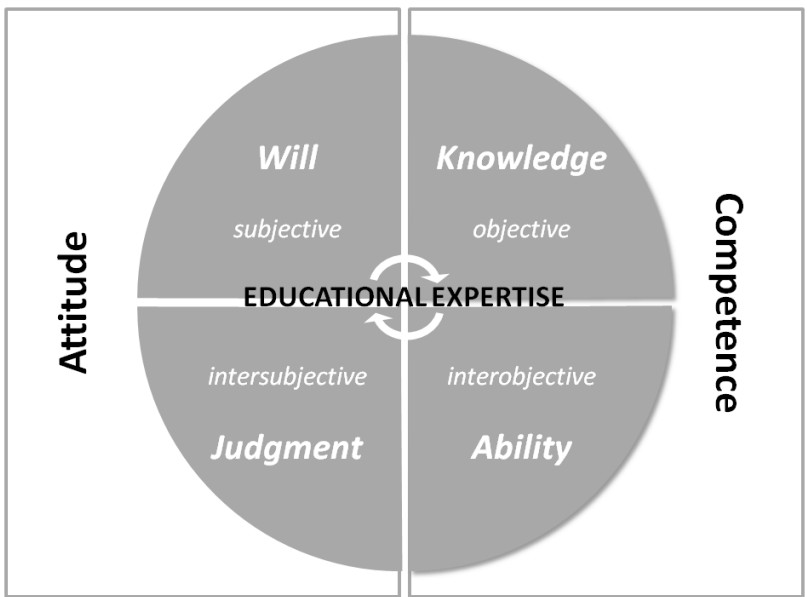

**Figure 3.** The ACAC model (attitude, competence, action, context) in relation to Ken Wilber's quadrant model.

## 4. Beliefs, Judgments, and Attitudes: Elements of Professional Attitudes

Although the argumentation presented above illustrates the difference between competence and attitudes in epistemological terms, the two nevertheless remain theoretical constructs that need to be explored in greater detail. While much progress has been made on this task for the concept of competence thanks to international comparative studies, as may be seen with the models of competence, it has yet to be tackled for the concept of attitudes. And yet it is no less important. Since attitudes are also referred to as "beliefs" and "judgments" in everyday language, it comes as no surprise that this multitude of terms also appears in the scientific discourse, leading to a lack of clarity. This raises the question as to whether or not the concept of attitudes is suitable for breaking down this mishmash of terms [21].

Here again, the already presented quadrant model by Ken Wilber provides an epistemological basis for more precisely defining the concept of attitudes, because it claims to enable a differentiated consideration of complex phenomena. Whereas we drew on the model in the previous step to illustrate that educational professionalism consists of competence and attitudes, the task is now to examine the concept of attitudes itself as a complex phenomenon. This way of going about things might at first glance seem repetitive or circular, but a closer look reveals that there are different processes of understanding and, thus, also different levels of observation at work here, which do require a similar methodological approach, yet have a different focus with regard to their content—doubtlessly a strength of Ken Wilber's quadrant model. With this in mind, how can we make the concept of attitudes more concrete? If we work on the assumption of a concept of attitudes consisting essentially of will and judgment, we can demonstrate with the help of Ken Wilber's epistemological position that this involves different grounds and validity claims (see Figure 4).

In the subjective quadrant, attitudes appear in the shape of desires, needs, and interests motivated primarily from a first-person perspective. They are, hence, "my" desires, "my" needs, and "my" interests. The validity claim connected with these forms of will and judgment is limited to the individual and cannot simply be applied unchanged to other people. This also includes expressions of belief, as already confirmed by the statement "I believe" on its own. From a conceptual standpoint, attitudes therefore become *subjective beliefs*, as articulated in statements like "I think that you love me."

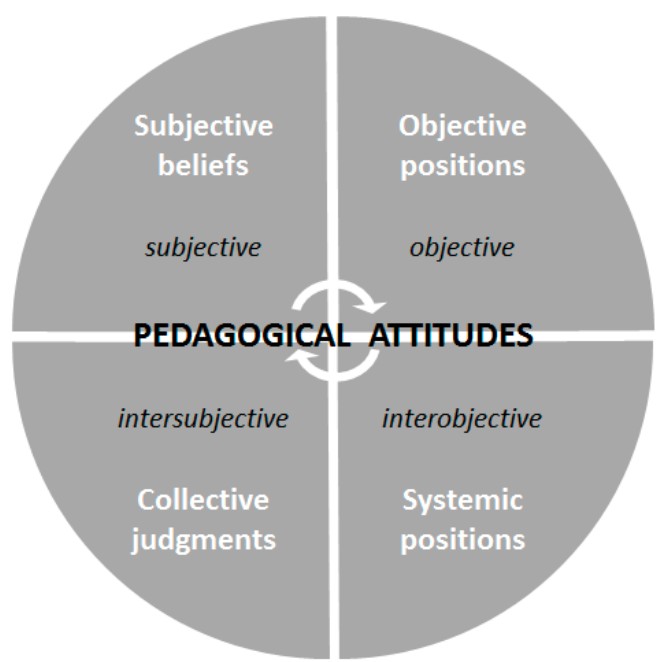

**Figure 4.** Mindframes in relation to Ken Wilber's quadrant model.

In the intersubjective quadrant, attitudes take on the form of values, norms, rules, and rituals. The special thing about these forms of will and judgment is that they are not formulated only by the individual, but are based on a collective. The collective agrees upon these values, norms, rules, and rituals in a process of exchange and discourse, and they are then valid not just for the individual, but for all people who regard themselves as belonging to this group. From a conceptual standpoint, attitudes therefore become *collective judgments* in this case, as articulated in statements like "We believe that human dignity is inviolable."

Finally, attitudes also appear in the objective and the interobjective context, both of which are determined by empirical methods and are, thus, considered together here, as forms of will and judgment that can be verified with the help of measurements and tests. Their claim to validity is, thus, detached from the individual, as well as from the group. From a conceptual standpoint, attitudes here become *objective or systemic positions*, as reflected in statements like "On the basis of empirical studies, I am of the opinion that the reduction of class sizes can only be effective if teachers change their teaching" [21].

Against the backdrop of this argumentation, the concept of attitudes takes on a more differentiated meaning: Attitudes can—depending on what experiences they appeal to—become beliefs and judgments. If attitudes are based on a subjective context, they are essentially subjective beliefs. If, on the other hand, they are based on an intersubjective context, they may be described primarily as collective judgments. And if they are based on an objective or interobjective context, they take on the form of *objective or systemic beliefs*. Experiences, thus, form the basis for attitudes and lead to particular forms of will and judgment, in particular to beliefs and judgments.

Each of these forms of will and judgment can have an impact on people's thoughts and actions and, thus, also on particular attitudes, on what people think and do in certain contexts. Competence in the form of knowledge and ability therefore depends on attitudes (cf. Figure 5). These attitudes and competences are always interdependent, together making up educational expertise in the form of mindframes (see Figure 5). While competences might be extensively discussed in literature (e.g., [22–24]), the practical use of the word is reduced to the areas of knowledge and ability. In this article, it is not possible to eliminate this discrepancy, although we are convinced that by distinguishing between competence and attitudes, it is possible to make the term more clear-cut. Further research on this is necessary.

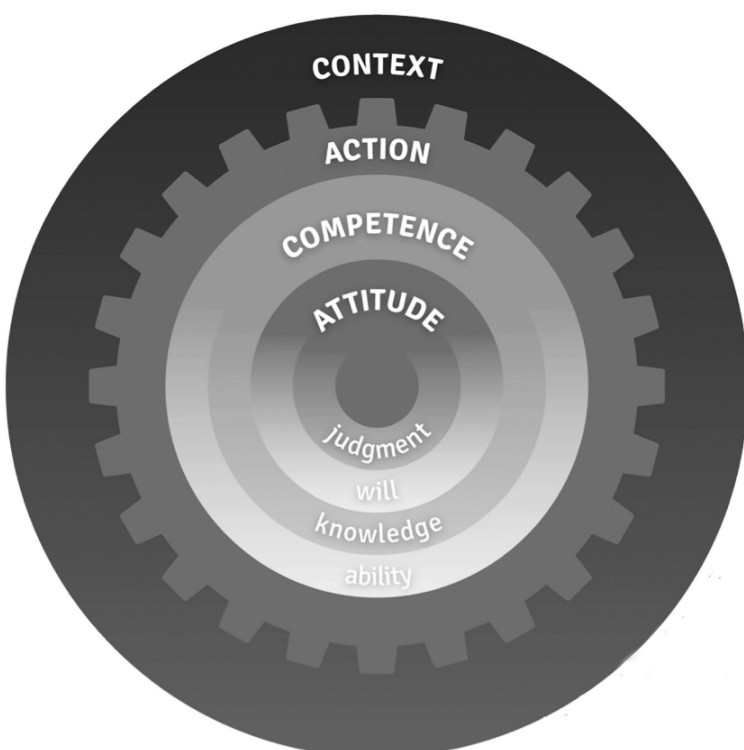

**Figure 5.** ACAC model of competence and attitude.

With regard to a professionalization of teacher education, it is important that we increasingly succeed in basing thought and action about school and instruction on attitudes. Professionalism is characterized not by beliefs in the form of simple expressions of belief or judgments handed down without further reflection, but rather by an appeal to evidence, as formulated by John Hattie [4]. Teacher professionalism is, thus, characterized by the fact that in regard to their mindframes, teachers possess corresponding competences and attitudes. August Hermann Niemeyer, one of the founding fathers of pedagogy as a university discipline and science, expressed this notion as early as 1813 in a letter to Johann Friedrich Herbart: "In this field I can be certain of the triumph of empiricism over speculation, without wishing in the slightest to take away from the latter its true worth in the right place" [1] (p. 180). It would, thus, be reductive to ignore or even push aside entirely the other facets of the concept of attitudes, seeing as they too can become effective and often contradict one another. The conclusion that should be drawn from this for teacher professionalism is not that the construct "attitudes" are a consistent form of will and judgment. It is rather a question of resolving the associated internal tensions to produce a coherent form of will and judgment on the basis of empirical evidence.

## 5. Mindframe Development: On the Measurability and Changeability of Mindframes in Teacher Education

The starting point of this paper was the observation that the current discourse on teacher professionalism in educational science takes insufficient account of Johann Friedrich Herbart's insight that a person's worth lies not only in knowledge and ability, but also, and especially, in will and judgment and that these modes of being of human existence also interact in many ways. In the argumentation presented thereafter, we substantiated this initial hypothesis and attempted to develop an alternative perspective on educational expertise with the ACAC model.

We are applying and studying this model in several projects within the context of the continuing education program "Visible Learning in Practice" at the University of Augsburg's Department of School Pedagogy. Following an initial pilot study, in which the questionnaire instrument was tested on teacher education students, we conducted several questionnaire studies on novice, advanced, and

expert teachers participating in continuing education programs on mindframe development in projects like "ProfiLe—Professionelle Lehrerrolle evidenzbasiert entwickeln" ("evidence-based development of teachers expertise") in cooperation with the Stiftung Bildungspakt Bayern (Bavarian Educational Foundation), "Strecke deine Hand aus" ("extend your hand") in cooperation with the school board of the Diocese of Augsburg, and "Schulen zum Leben" ("schools for life") in cooperation with the government of the Land of Mecklenburg-Vorpommern [25,26]. These projects also aim to reveal the course of, as well as changes in, competence and attitudes, and consequently in ability, knowledge, will, and judgment.

In the following, we would like to provide an outline of the project "schools for life" as an example and present initial findings of the evaluation: The aim of this school development project in Mecklenburg-Vorpommern is to gain insight on what can be described as educational expertise. At the same time, however, the findings can provide important indications on how to become a successful teacher in the first place and how to remain one throughout one's life. The "schools for life" project, thus, focuses on developing competence and attitudes and endeavors, on the basis of the ten mindframes from "Visible Learning" listed above, to integrate them into everyday school and teaching practice. The courses treat one mindframe each. As a means of further scrutinizing their own attitudes and identifying possible steps in their development as teachers, the teachers then take a look at research. In the subsequent working phase, they work out the meaning of the mindframe and consider resulting possibilities for action in their own teaching. The purpose of reflecting and working on the mindframes is to develop not just competence, but also attitudes. The development of attitudes is brought about by intensive reflection. The teachers now think differently about things, try out new things, and have also changed their behavior accordingly. The main aim of the three-year project, in which a comprehensive school and two academic-track secondary schools in Mecklenburg-Vorpommern are participating, is to reflect on and develop the educational expertise of teachers. The program engages the entire teaching staff in a dialogue, in which they reflect on their own professionalism and receive support in developing school and instruction.

To gain knowledge on teachers' educational expertise, we are collecting empirical data parallel to the further training modules and instruction. Each mindframe has its own brief questionnaire. They contain 8 to 12 items and collect data on ability, will, knowledge, and judgement with the help of the incomplete sentences (see Table 1: "I am very good at . . . ", "I know perfectly well . . . ", "My goal is always . . . " and "I believe that . . . "). The teachers ($N_1$ = 102; $N_2$ = 73) assess their own ability, knowledge, will, and judgments in the classroom in relation to the ten mindframes. The facets of ACAC are covered by various incomplete sentences, as the example of the mindframe "I set the challenge" shows.

**Table 1.** Sample items of the teacher self-assessment questionnaire (scale: "I set the challenge").

| | |
|---|---|
| Ability | I am very good at developing challenging assignments on the basis of learning levels. |
| | I am very good at setting challenging learning goals on the basis of learning levels. |
| Knowledge | I know perfectly well that the tasks in my lessons need to be challenging. |
| | I know perfectly well that the learning requirements need to be challenging for the students. |
| Will | My goal is always to design my lesson to include challenging goals based on the learning level. |
| | My goal is always to design assignments to be challenging for the learners. |
| Judgment | I believe that it is important to see to it that learners need to make an effort. |
| | I believe that it is important to see to it that the learners are challenged. |

Ability is represented by "I am very good at . . . ," knowledge by "I know perfectly well . . . ," will by "My goal is always to . . . ," and judgment by "I believe that . . . " The completion of the sentence takes up the specific content of the mindframe "I set the challenge." The teachers rate their own mindframes on a four-point Likert scale from "I don't agree at all" to "I fully agree."

As we already had a clear idea of the possible factors, we conducted a confirmatory factor analysis to validate the questionnaires. On the basis of the ACAC model, we assumed four factors for each

of the ten questionnaires. The model fits were acceptable to good. All factors of the questionnaire achieved good reliability scores between $\alpha = 0.81$ and $\alpha = 0.92$. The questionnaire was completed before and again some time after the respective continuing education course, allowing us to measure changes over time. We measured the development of ability, knowledge, will, and judgment with regard to the 10 mindframes with the help of the Wilcoxon test and the calculation of effect sizes. The initial longitudinal results (pre-test in November 2016 and follow-up survey in June 2018) showed a positive development in all four facets (see Figure 6). We may therefore conclude that the continuing education courses led not just to a significant increase in competence on the part of the teachers, but also to a significant change in their mindframes.

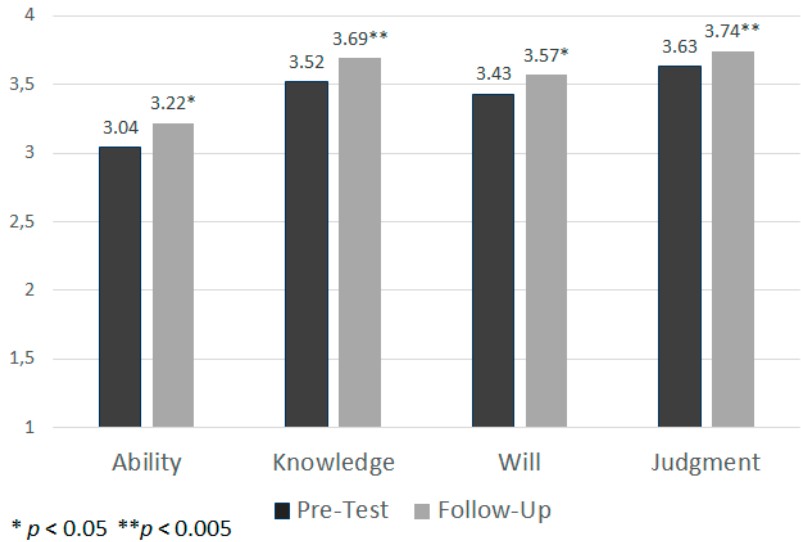

**Figure 6.** Teacher self-assessment: Longitudinal data (means).

The series of continuing education courses had an overall positive effect of $d = 0.31$ on the teachers' ability. Their knowledge increased to an effect size of $d = 0.32$. The teachers' will achieved an effect size of $d = 0.26$ and their judgment an effect size of $d = 0.23$. This rough picture of various facets of educational expertise becomes more specific when one takes a closer look at the results on several mindframes. In this paper, we would like to select "I talk about learning, not about teaching," "I set the challenge," and "I see learning as hard work" as examples, because these mindframes show clear results: The means of the self-assessment regarding ability, knowledge, will, and judgment from the pre-test to the follow-up survey show an increase in all points, half of them constituting significant changes (cf. Table 2).

**Table 2.** Data from the teacher survey by means of the self-assessment questionnaire.

| | | Pre-Test (N₁ = 102) | | Follow-Up (N₂ = 73) | | | |
|---|---|---|---|---|---|---|---|
| | | **MD** | **SD** | **MD** | **SD** | **d** | **α** |
| **Mindframe 1:** "I focus on learning and the language of learning." | ability | 3.00 | 0.54 | 3.08 | 0.51 | | |
| | knowledge | 3.25 | 0.45 | 3.63 ** | 0.49 | 0.81 | |
| | will | 3.31 | 0.59 | 3.52 * | 0.55 | 0.37 | 0.84 |
| | judgment | 3.73 | 0.41 | 3.79 | 0.36 | | |
| **Mindframe 2:** "I strive for challenge and not merely 'doing your best'." | ability | 2.80 | 0.63 | 3.00 * | 0.64 | 0.31 | |
| | knowledge | 3.45 | 0.55 | 3.68 * | 0.47 | 0.44 | |
| | will | 3.32 | 0.55 | 3.49 | 0.60 | | 0.85 |
| | judgment | 3.46 | 0.53 | 3.75 ** | 0.42 | 0.60 | |
| **Mindframe 3:** "I recognize that learning is hard work." | ability | 3.21 | 0.52 | 3.27 | 0.54 | | |
| | knowledge | 3.83 | 0.33 | 3.90 | 0.27 | | |
| | will | 3.54 | 0.48 | 3.62 | 0.47 | | 0.81 |
| * $p < 0.05$, ** $p < 0.005$ | judgment | 3.62 | 0.46 | 3.84 ** | 0.31 | 0.54 | |

The effect sizes obtained in this study demonstrate that it is possible to achieve an increase in competence (mindframe 1: Knowledge at d = 0.81 and mindframe 2: Knowledge at d = 0.44) as well as a change in mindframes (mindframe 2: Judgment at d = 0.60 and mindframe 3: Judgment at d = 0.54) within the context of continuing education courses. The crucial point for this paper is that they also confirm the validity of a construct of educational expertise focusing on competence and mindframes.

## 6. Conclusion

As stated at the beginning of this paper, Johann Friedrich Herbart emphasized the interrelationship between knowledge and will as early as 1835. Studies like John Hattie's synthesis have brought the significance of judgment and will back into the focus of empirical educational research. Teacher professionalism involves more than just subject matter knowledge, pedagogical competence, and didactic competence. Rather, the effectiveness of this triad depends above all on the teacher's mindframes.

The results of our studies are subject to several limitations. First of all, the sample was very small, which has consequences for the validity both of the confirmatory factor analysis and of the results. Another limitation has to do with the complexity of the underlying construct. The measurement instrument is supposed to measure teacher professionalism both in the form of competence and attitudes and with regard to the 10 mindframes. As a means of verifying this complexity, a further survey including more test subjects should be conducted. Another question that should be addressed is that of program desirability to determine the extent to which there has actually been a lasting change in teacher professionalism.

Even though the study has limitations in these respects, the results showed that teacher expertise can be depicted with the help of the ACAC model and that the continuing education courses have a positive effect on the professionalism of the participating teachers.

If we wish to change something and improve ourselves, we should begin by becoming aware of our attitudes and scrutinizing our own teaching. An example of this is reflection by means of feedback: Why do we give feedback, and why should we seek feedback? An exchange with colleagues and learners can be helpful. The important thing is to avoid rushing things. Changes take time and are a process. Initiating this process is definitely as difficult as bringing it to a successful conclusion.

**Author Contributions:** Conceptualization, K.Z.; Formal analysis, K.Z., C.L., J.T. and D.W.; Investigation, K.Z.; Project administration, K.Z.; Supervision, K.Z.; Validation, C.L. and D.W.; Visualization, C.L. and J.T., Writing—original draft, K.Z.; Writing—review and editing, C.L. and J.T.

**Funding:** This research received no external funding.

**Conflicts of Interest:** The authors declare no conflict of interest. The founding sponsors had no role in the design of the study; in the collection, analyses, or interpretation of data; in the writing of the manuscript, and in the decision to publish the results.

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
