# Peer review of "Teacher Mindframes from an Educational Science Perspective"

_education, doi:10.3390/educsci8040209_

Round 1

Reviewer 1 Report

This manuscript addresses the importance of mindframes, together with (sic) competence, in teachers’ professionalism. Firstly, the authors argue that educational expertise relies not only on knowledge and ability, but also on attitudes and values, which are referred to as “mindframes”. The authors propose a new, synthetic model, the KW3 model, to conceptualiza and measure mindframes. The model is being applied in several pilot studies, one of which is reported, as an example, in the paper.

The paper is nicely structured and very well writen. The arguments within are well supported with relevant literature.

I have, however, three main concerns, regarding:

- The use of the Word "competence" for referring to knowledge and ability

- The seemingly doublé meaning of "mindframes" in the text, which could be related with the use of que quadrant model to interpret two different ideas.

- The statistical procedure, which is poorly described.

Please see the attached document for further details

Author Response

thank you very much for your detailed review.

response to point 1:

We found it to be very valuable, especially your remarks on imprecisive terminology concerning the expressions "mindframes" and "attitudes". This was due to difficulties which arose when we had to translate the text - however the imprecisions have been taken care of and more clear-cut terms are used now.

response to point 2:

The statistical procedure was revised and the description of the sample as well as the validation of the questionnaires were added.

response to point 3:

We define "competences" as a component of pedagogical expertise, which can include both factual and procedural knowledge as well as the capacity to solve complex tasks.

Thank you, too, for the "minor concerns", which were very useful and step-by-step integrated in our revised version.

We hope that the new version now reflects the changes suggested in your review.

Kind regards

Authors

Reviewer 2 Report

This is a well written paper which creates an interesting juztaposition between a phisophical framework, a pedagogical framework, and then applies this sythesis to a question of teacher efficancy.The nature of a tecaher's knowledge and skills remains a pertinent subject for study.

There are three points which need addressing.

The first is minor; although will is described in relation to other concepts and attributes, it is not clearly defined in one sentence, which would really help clarify the piece for readers.

The second point is more significant.The assumption that knowledge makes a tecaher effetcive, countered by Hattie's meta-analysis showing that it doesn't, is a very simple analysis of the role of subject knowledge. I think Shulman's notion of pedagogic content knowledge may have a contribution to make to this analysis. For many prectitioners and researchers, the question is rather 'what is the minmum level fo knowledge needed for effectiveness'. This is evident in England's setting up of 'top up' courses in shortage areas (Subject Knowledge Enhancement courses). This over-simplification at the start of the paper detracts from the analysis.

The third point is major. The questions are very leadingand can only be expected to give biased responses. Whilst I appreciate that some meaning may have been lost in translation, the questions are one-sided. eg 'I focus on the langauage of learning' invites a positive answer. What was (I suspect) wanted was to find find out where the balance lay between the general language of learning and the specific subject content of the lesson, but this wasn't expressed in the questions. Other questions, even when they were not requiring a choice between two options,  were very loaded and were, I think, unlikely to produce a full range of responses. Because of this flaw, I think the findings cannot be considered valid.That still leaves a philosophical analysis whihc merits publication, however.

Author Response

thank you very much for your review.

Point 1: You are right that "will" is not explicitely defined. However, the term is integrated in the broader concept of the ACAC model and we hope its meaning is clearly anchored in this context. We did indeed sharpen our definitions of "pedagogical expertise" (attitude & competence) and we hope that because of these revisions the terminology as a whole becomes more clear-cut.

Point 2: Thank you very much for your remarks concerning the importance of the "subject matter knowledge" of teachers. We agree that this it is a controversely discussed statement that Hattie makes at this point. It would definitely need further discussion and an exchange of arguments for and against it.

Point 3: You are absolutely right that the mindframes are not questions but statements that define a set of attitudes and competencies. The empirical findings were indeed not complete in the first version, however further data has been provided and the descriptions have been made more clear.

Kind regards

Authors

Reviewer 3 Report

I think the article should be structured on a template: introduction, methods, results, discussions and conclusions. More information about research subjects (age, gender, work experience). Which statistical method applied to the questionnaire? The structure of table 2 needs to be rebuilt because the data in the table is not understood. Translation of projects from German into English. Insufficiently developed discussion and conclusions.

Author Response

thank you very much for your feedback. We did include more information about research subjects, as you suggested.

The structure of table 2 was rebuilt. We also translated the projects from German into English and included it in brackets after the German name.

We revised and extended the discussion and the conclusion.

Thank you again for all your helpful remarks.

Kind regards,

Authors

Round 2

Reviewer 1 Report

Now that some terms have been defined with more specificity, the manuscript looks sounder and more transparent. I have enjoyed myself reading this second version as much as I did with the previous one.

I find still disputable the use of "competence" to refer just to knowledge and abilities (put into action). It is widely agreed that the notion of competence also encompasses wills and judgments, which you consider under "attitudes." I see your point entirely, but I would suggest you use another word, instead of competence.

At least once the word competency is used. Competency and competence may be understood to refer to different concepts; if this is intentional, please explain when do you use each of the terms, and their respective meaning; if it is an error, I would suggest you homogenize the wording. 

I have still some doubts about the statistical procedure, which I must admit has been described much better. For example, the alpha values for each of the factors (=facets of ACCA) have been calculated for each mindframe or across the 10... Probably all those details would add little to the manuscript, but they could maybe be provided as supplementary material, for the reader to judge by him/herself the reliability of the results.

Thank you again for this nice, though-provoking piece of work.

Author Response

Thank you very much for this second, very in-depth review.

Answering your remarks:

While competences might be extensively discussed in literature, the practical use of the word is reduced to the areas of knowledge and ability. In this article it is not possible to eliminate this discrepancy, although we are convinced that by distinguishing between competence and attitudes it is possible to make the term more clear-cut. Further research on this is necessary.

--> We refer to this point with a footnote in the text on page 8.

The imprecision with regards to the expressions of "competence" and "competency" has been corrected.

In order to make our "Statistical prodecure" more clear, we have provided supplementary material ("α values for the teacher questionnaire") which is attached to the essay. It includes the Alpha-values for the factors as well as for the ACAC-model.

Kind regards,

Christina Lachner & Jonas Tögel

Reviewer 2 Report

The revisions that have been undertaken have made the paper clearer, thank-you. I think that you could develop the consideration of the precise impact of subject knowledge on teacher effectiveness in future work very usefully, were you minded to do so.

Author Response

thank you very much for your positive feedback.

Kind regards

Christina Lachner